# MAGTRACK: MLLM-AUGMENTED GROUNDING AND TEXT REFINEMENT FOR LANGUAGE-GUIDED TRACKING

## ABSTRACT

Language-guided object tracking aims to locate the target in a video based solely on a natural language description, without any bounding box supervision. While recent methods have made encouraging progress by incorporating language into visual tracking, most treat it as an auxiliary signal rather than a primary driver. This limits their effectiveness in fully language-only scenarios, which remain underexplored despite their user-friendly nature. In this paper, we propose MAGTrack, a novel framework for language-guided object tracking that seamlessly integrates Multimodal Large Language Models (MLLMs) without requiring additional training. MAGTrack tackles key challenges through two plug-and-play modules: the MLLM-based Grounding Module (MGM) and the MLLM-based Text Refinement Module (TRM). MGM leverages MLLM reasoning to achieve accurate initial target localization, even in challenging scenarios with visually similar objects. Complementarily, TRM dynamically updates the textual description based on the current visual context and tracking history. Extensive experiments on four benchmarks—OTB99, TNL2K, LaSOT, and LaSOT$_{ext}$—demonstrate that MAGTrack consistently improves both first-frame grounding and long-term tracking accuracy, achieving state-of-the-art performance under the language-only setting.

## 1 INTRODUCTION

Language-guided object tracking (Li et al., 2017) aims to locate the target in a video based solely on a natural language description provided in the first frame, without relying on any bounding box supervision. Compared to conventional tracking methods (Wei et al., 2023; Bai et al., 2024; Ye et al., 2022; Chen et al., 2023) that depend on manually annotated boxes, this paradigm eliminates explicit spatial input, offering a more challenging yet natural form of human-computer interaction. By replacing box-based initialization with language, this setting enables greater practicality and scalability in real-world applications such as robotics, surveillance, and augmented reality (MacKenzie, 2024; Haresh et al., 2024).

Although recent vision-language tracking methods (Wang et al., 2021; Li et al., 2022b) have shown promising results using natural language descriptions, their performance still falls short compared to approaches that rely on manually annotated bounding boxes—particularly in the absence of spatial supervision. As illustrated in Figure 1(a), language-only trackers commonly face two major challenges: **1) Uncertainty in First-Frame Localization.** Without spatial priors, the model must infer the target's identity and location solely from the description, which can lead to confusion among visually similar objects. **2) Temporal Misalignment of Static Descriptions.** Most methods rely on a fixed initial description throughout the sequence, failing to adapt to changes in the target's appearance or context, resulting in a growing mismatch between language and visual content. While some existing approaches (Zhou et al., 2023; Ma et al., 2024) incorporate language as an auxiliary signal to enhance visual features, they typically focus on unified architectures or multimodal fusion strategies, rather than addressing the unique challenges of the language-only setting. As a result, they struggle to reason effectively in ambiguous or dynamic scenes, especially when the language description is vague or becomes outdated.

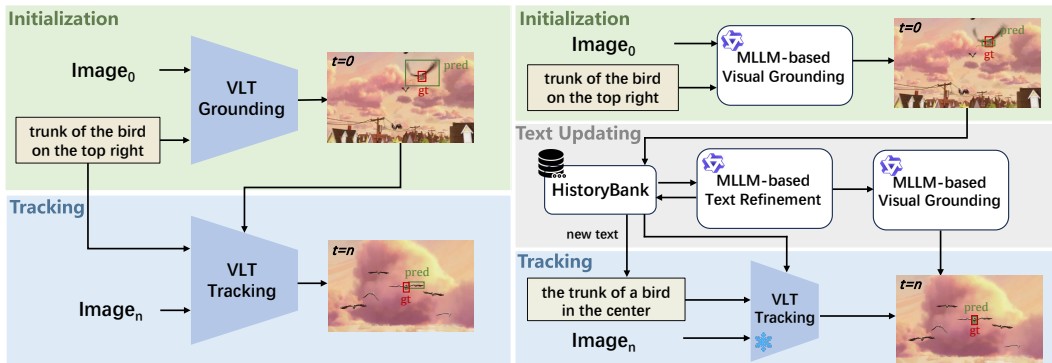

(a) Conventional language-guided tracking progress      (b) Our MLLM-augmented tracking pipeline

Figure 1: Comparison between conventional language-only tracking and our proposed framework. (a) Conventional methods rely solely on the initial natural language description throughout the process. (b) Our method incorporates MLLM-based visual grounding and text refinement modules, which improve target localization by leveraging historical cues and dynamically updated textual descriptions.

In recent years, Multimodal Large Language Models (MLLMs) (Wang et al., 2024a; Touvron et al., 2023; Wang et al., 2024b; Zhu et al., 2023) have emerged as powerful tools for vision-language tasks, exhibiting strong generalization and reasoning capabilities (Chen et al., 2024; Zhao et al., 2025; Pan et al., 2024). While several recent works have explored the use of MLLMs in visual tracking, their applications typically rely on additional training or ground-truth bounding box supervision. For example, Liu et al. (2024b) perform real-time language updates via MLLMs but require task-specific fine-tuning, whereas ChatTracker (Sun et al., 2024) leverages MLLMs for prompt refinement based on manually annotated boxes—resources that are unavailable in language-only settings. As a result, the potential of MLLMs in the language-only setting remains largely unexplored.

To address the challenges in the language-only setting, we propose **MAGTrack**, a flexible framework with two **plug-and-play modules**, as shown in Figure 1(b): the **MLLM-based Grounding Module (MGM)** and the **MLLM-based Text Refinement Module (TRM)**. Notably, both of these modules are operational solely during inference. MGM combines a grounded vision-language module (GVLM) with the reasoning power of an MLLM to achieve accurate and robust localization, even under heavy distractor interference from semantically similar objects. Building on this foundation, MGM focuses on accurate target initialization, while long-term tracking also demands adaptability to appearance changes and contextual variations. To address this, we introduce TRM, which continuously updates the initial natural language description using current visual cues and tracking history. This dynamic refinement helps maintain alignment between language and visual content throughout the sequence. We validate our approach across four standard benchmarks—OTB99 (Li et al., 2017), TNL2K (Wang et al., 2021), LaSOT (Fan et al., 2019), and LaSOT$_{ext}$ (Fan et al., 2021)—where MAGTrack consistently improves both grounding accuracy and tracking robustness under the language-only setting.

Our main contributions are summarized as follows:

1. We present MAGTrack, a modular framework that integrates MLLMs into language-guided object tracking, enabling effective inference without additional training or spatial annotations.

2. We introduce two plug-and-play modules: MGM enhances first-frame localization by leveraging MLLM reasoning to handle semantically similar distractors, while TRM refines the initial language description over time to stay aligned with dynamic visual content.

3. Experiments on four standard benchmarks show that MAGTrack achieves state-of-the-art performance under the language-only setting, demonstrating its potential for more natural and interactive human-AI interfaces in future tracking systems.

## 2    RELATED WORK

### 2.1    LANGUAGE-GUIDED OBJECT TRACKING

Vision-language tracking (VLT) incorporates language into object tracking to enhance tracking accuracy. While many methods use language to assist box-initialized tracking (Zhang et al., 2023; Guo et al., 2022; Li et al., 2023; Zheng et al., 2024), we focus on the language-guided object tracking (Li et al., 2017; Zhou et al., 2023; Ma et al., 2024), which is more challenging due to the lack of spatial supervision and evolving visual content. Several datasets with rich textual annotations (Wang et al., 2021; Fan et al., 2019; Hu et al., 2023), provide a foundation for research in this direction. In the language-only setting,  Li et al. (2017) first formalize the task and demonstrate its potential for semantic-level tracking. Early works such as  Yang et al. (2020b) and  Feng et al. (2020) decompose the task into separate grounding and tracking stages. However, these methods lack sufficient cross-modal interaction, resulting in unstable performance. Subsequently, JointNLT (Zhou et al., 2023) reformulates grounding and tracking as a unified task, introducing an end-to-end framework for localizing language-referred targets. UVLTrack (Ma et al., 2024) extends this idea with a contrastive learning framework that supports both language and visual references within a single model. Despite their effectiveness, these methods assume a static language query throughout tracking and fail to adapt to the target's appearance changes. To address this, QueryNLT (Shao et al., 2024) introduces a query refinement module that identifies and removes redundant phrases from the original description. Although it considers the issue of semantic mismatch, its refinement is limited to filtering and lacks the ability to generate updated or context-aware language.  MemVLT (Feng et al., 2024) further explores prompt adaptation through memory-augmented modelling inspired by Complementary Learning Systems (CLS) theory, enabling temporal flexibility in multimodal prompting. However, this method requires task-specific training and is constrained by a fixed architecture. In contrast, our method leverages pretrained multimodal large language models (MLLMs) to perform both first-frame grounding and dynamic language refinement entirely at inference time. Crucially, it enables the generation of new, context-aware descriptions, allowing the tracker to adapt to appearance shifts without retraining.

### 2.2    MULTIMODAL LARGE LANGUAGE MODELS IN TRACKING

Multimodal large language models (MLLMs) (Wang et al., 2024a; Chen et al., 2024; Liu et al., 2023; Li et al., 2022a; Xu et al., 2023) extend the capabilities of large language models (LLMs) (Achiam et al., 2023; Team et al., 2023; Touvron et al., 2023) by incorporating visual inputs, enabling joint reasoning over language and vision. These models have achieved strong performance across a wide range of tasks, including anomaly detection (Zanella et al., 2024), video understanding (Tang et al., 2025), and medical image analysis (Kim et al., 2024). Recently, MLLMs have also been explored in the context of object tracking. DTLLM-VLT (Li et al., 2024) utilizes MLLMs to automatically generate multi-granularity textual annotations, improving tracking benchmarks through enhanced semantic diversity. At the algorithmic level,  Liu et al. (2024b) propose a real-time updating framework that continuously refines language descriptions based on visual observations. While effective, this method relies on additional training on tracking datasets, which limits its generalizability to unseen domains. ChatTracker (Sun et al., 2024) introduces a reflection-based prompt optimization (RPO) module that iteratively improves ambiguous or inaccurate queries using feedback from tracking results. However, the RPO module is conditioned on ground-truth bounding boxes to guide prompt reflection, making the method dependent on annotated spatial supervision and unsuitable for language-only tracking settings. While prior works demonstrate the potential of MLLMs in visual tracking, they either depend on task-specific training or require annotated spatial supervision to function effectively. To overcome these limitations, we propose a general-purpose framework that fully exploits the reasoning and generative capabilities of MLLMs for adaptive language-guided tracking in a training-free and annotation-free manner.

## 3    METHOD

In this section, we present MAGTrack, a framework for language-only object tracking that requires no additional training. It integrates multimodal large language models (MLLMs) through two plug-and-play modules: a grounding module (see Sec. 3.3) that identifies the target in the initial frame

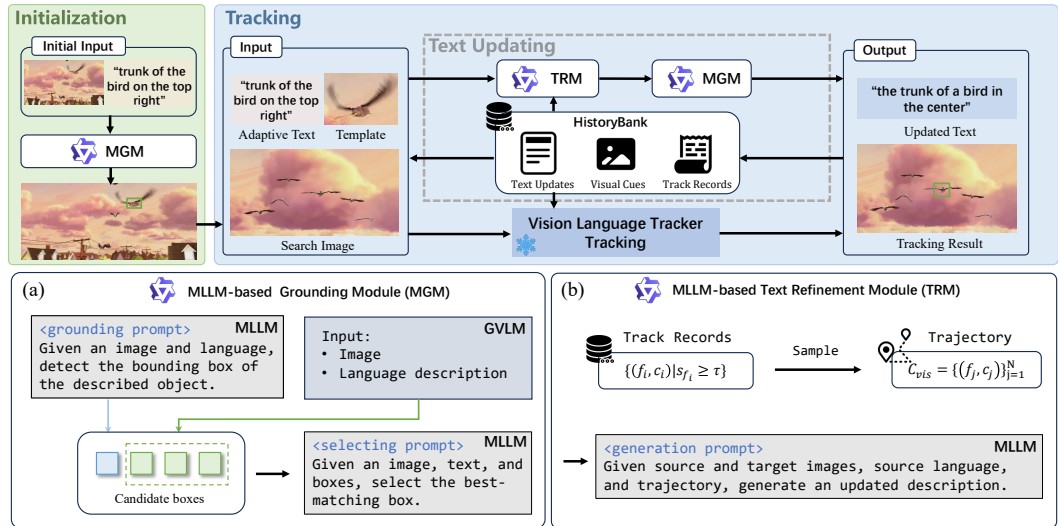

Figure 2: MLLM-augmented language-guided tracking framework. Our approach enhances a Vision-Language Tracker (VLT) by incorporating two plug-and-play modules: (a) the MLLM-based Grounding Module (MGM) leverages prompt-based reasoning for enhanced localization, and (b) the MLLM-based Text Refinement Module (TRM) dynamically refines the language description using historical visual information and tracking trajectories. The integrated framework facilitates robust and adaptive target tracking in challenging ambiguous or dynamic environments.

without bounding box input, and a text refinement module (see Sec. 3.4) that updates the query based on evolving visual context. The entire system operates purely at inference time and does not rely on any spatial supervision.

## 3.1 PRELIMINARIES

**Task Formulation.** Given a video $V = \{I_t\}_{t=0}^{T}$ with $T$ frames and a natural language description $L = \{w_1, w_2, \ldots, w_N\}$ provided for the first frame $I_0$, the goal of language-guided object tracking is to predict a sequence of object positions $\{P_t\}_{t=0}^{T}$ without relying on any bounding box annotations, where each $P_t$ denotes the estimated location in frame $I_t$.

**Model Components.** We denote the Multimodal Large Language Model (MLLM) as a function $\mathcal{M}$ that performs cross-modal reasoning over an image and a language prompt. Given an image $I$ and a textual `prompt`, the MLLM produces a natural language response:

$$R = \mathcal{M}(I, \texttt{prompt}), \tag{1}$$

where $R$ is a plain-text output whose interpretation depends on the prompt design. Depending on the task, $R$ can be post-processed into various forms, such as a bounding box prediction, a refined description, or a semantic decision.

We define the Grounded Vision-Language Model (GVLM) as a function $\mathcal{G}$ that generates language-conditioned object proposals. Given an image $I$ and the natural language description $L$, the GVLM outputs a set of candidate bounding boxes with associated confidence scores:

$$\{b_i\}_{i=1}^{K} = \mathcal{G}(I, L), \tag{2}$$

where $b_i \in \mathbb{R}^4$ is the $i$-th bounding box and $K$ is the total number of proposals. We adopt Grounding DINO (Liu et al., 2024a) as the GVLM in our implementation.

The Vision-Language Tracker (VLT) is defined as a function $\mathcal{T}$ that performs frame-wise localization guided by both language and a visual template. Given a search frame $I_t$, a fixed reference `template` extracted from the initial localization, and the language description $L$, the tracker predicts the target position and a confidence score:

$$(P_t, s_t) = \mathcal{T}(I_t, \texttt{template}, L), \tag{3}$$

where $P_t \in \mathbb{R}^4$ denotes the predicted bounding box in frame $I_t$, and $s_t \in [0, 1]$ reflects the model's confidence. The tracker is applied to each frame $t > 0$ using the template from frame $I_0$.

## 3.2 FRAMEWORK OVERVIEW

As illustrated in Figure 2, our framework integrates multimodal reasoning and adaptive tracking to address the language-only tracking setting. Given a video sequence, we first use an MLLM-based grounding module to infer the initial object location $P_0$ from the first frame $I_0$ and the language description. This initial position serves as the `template` for a vision-language tracker, which then performs frame-by-frame prediction. For each $t > 0$, the tracker produces a bounding box $P_t$ and an associated confidence score $s_t$, following $(P_t, s_t) = \mathcal{T}(I_t, \texttt{template}, L)$. To detect potential tracking failures, we monitor $s_t$ over time. Once the confidence drops below a predefined threshold $\tau$ for $k$ consecutive frames, i.e., $s_{t-k+1}, \ldots, s_t < \tau$, we invoke a text refinement module that leverages the generative capabilities of the MLLM to reassess and optionally update the original description. The refined query, denoted as $L_{new}$, is then used to assist the tracker in re-localizing the target, allowing the system to recover from drift or ambiguity. This design combines the complementary strengths of large-scale multimodal reasoning and efficient temporal tracking, enabling robust object localization under weak supervision.

## 3.3 MLLM-BASED GROUNDING MODULE

The MLLM-based Grounding Module plays a key role in initializing the tracker by identifying the target object in the first frame $I_0$ based solely on the natural language description $L$. While primarily designed for first-frame localization, this module can also be leveraged in subsequent stages to assist tracking under ambiguous or challenging conditions.

To perform grounding, we first leverage the intrinsic grounding capability of the MLLM itself using a $\texttt{prompt}_{\text{ground}}$, which directly produces a bounding box prediction:

$$b_{\text{MLLM}} = \mathcal{M}_{\text{ground}}(I_0, \texttt{prompt}_{\text{ground}}). \tag{4}$$

Inspired by the strategy proposed in (Sun et al., 2024), we apply a Grounded Vision-Language Model (GVLM) in parallel to generate a set of language-conditioned region proposals:

$$\{b_i\}_{i=1}^K = \mathcal{G}(I_0, L). \tag{5}$$

The direct output $b_{\text{MLLM}}$ and the proposals $\{b_i\}_{i=1}^K$ are then combined and encoded into a unified prompt, which is passed again to the MLLM using a $\texttt{prompt}_{\text{select}}$ instruction. The MLLM performs cross-modal reasoning over all candidate boxes and selects the one most semantically aligned with $L$:

$$P_0 = \mathcal{M}_{\text{select}}(I_0, \{b_{\text{MLLM}}\} \cup \{b_i\}_{i=1}^K, \texttt{prompt}_{\text{select}}). \tag{6}$$

This two-stage grounding strategy combines the high-quality region proposals from the GVLM (Liu et al., 2024a) with the strong reasoning capabilities of the MLLM, effectively mitigating the inherent ambiguity of natural language and visual distractors, and enabling our method to produce more accurate and reliable initializations under the language-only setting. Detailed experimental validation of this two-stage grounding strategy can be found in Sec. 4.3.

## 3.4 MLLM-BASED TEXT REFINEMENT MODULE

To enhance adaptability under appearance variations or tracking drift, we introduce a text refinement module that leverages the generative capabilities of MLLMs to produce updated descriptions tailored to the target's current state. To maintain tracking efficiency, the module is only activated when necessary. Specifically, it is triggered when the vision-language tracker exhibits persistently low confidence over a consecutive window of $k$ frames, i.e., $s_{t-k+1}, \ldots, s_t < \tau$, where $\tau$ is a predefined threshold.

Upon activation at frame $t$, the module gathers visual history from the tracker. Instead of using all high-confidence predictions, we construct a compact pseudo-trajectory by selecting representative points from the interval $[0, t]$. Let $\mathcal{B}_{\text{high}} = \{(f_i, c_i) \mid s_{f_i} \geq \tau\}$ be the set of center coordinates $c_i$ from

Table 1: State-of-the-art comparison of tracking methods under the language-only setting on four datasets. $\text{TEXT}_u$ indicates whether textual information was updated during tracking. Red and blue values represent the best and second-best scores, respectively. Entries marked with * are reproduced using official code in our evaluation environment. All values are reported as percentages.

| Method | $\text{TEXT}_u$ | OTB99 | | | TNL2K | | | LaSOT | | | LaSOT$_{ext}$ | | |
|---|---|---|---|---|---|---|---|---|---|---|---|---|---|
| | | AUC | P | $\text{P}_N$ | AUC | P | $\text{P}_N$ | AUC | P | $\text{P}_N$ | AUC | P | $\text{P}_N$ |
| GTI  (Yang et al., 2020a) | ✗ | 58.1 | - | 73.2 | - | - | - | 47.8 | - | 47.6 | - | - | - |
| TNL2K-I (Wang et al., 2021) | ✗ | 19.0 | - | 24.0 | 11.0 | 11.0 | 6.0 | 51.0 | - | 49.0 | - | - | - |
| CTRNLT (Li et al., 2022b) | ✗ | 53.0 | - | 72.0 | 14.0 | 15.0 | 9.0 | 52.0 | - | 51.0 | - | - | - |
| JointNLT*  (Zhou et al., 2023) | ✗ | 57.2 | 75.3 | 68.3 | 54.5 | 54.8 | 70.5 | 56.8 | 59.2 | 64.4 | 34.9 | 35.5 | 41.3 |
| UVLTrack-B*  (Ma et al., 2024) | ✗ | 60.1 | 76.9 | 71.1 | 54.9 | 56.6 | 71.0 | 56.7 | 60.1 | 64.1 | 30.6 | 31.6 | 36.1 |
| QueryNLT  (Shao et al., 2024) | ✓ | 61.2 | 81.0 | 73.9 | 53.3 | 53.0 | 70.4 | 54.2 | 55.0 | 62.5 | - | - | - |
| ATFUVLT-B  (Liu et al., 2024b) | ✓ | 60.9 | 78.1 | 72.1 | 54.9 | 56.7 | 71.2 | 57.1 | 60.8 | 64.6 | - | - | - |
| MAGTrack(Ours) | ✓ | 65.3 | 88.2 | 79.8 | 58.2 | 60.7 | 75.1 | 60.1 | 63.9 | 67.4 | 42.2 | 48.2 | 51.2 |

high-confidence frames $f_i$. We apply a stratified sampling function $\text{Sample}(\cdot)$ to obtain a fixed-length representative trajectory:

$$\mathcal{C}_{\text{vis}} = \text{Sample}(\mathcal{B}_{\text{high}}, t), \tag{7}$$

where $\mathcal{C}_{\text{vis}} = \{(f_j, c_j)\}_{j=1}^N$ includes $N$ sampled frame-center pairs. The sampling strategy divides the frame interval into early, mid, and late segments, prioritizing diversity and recency. The sampling strategy is detailed in Algorithm 1 in Appendix A.5.

With this trajectory, the MLLM is prompted using: (1) the initial frame $I_0$ and language $L$, (2) the current frame $I_t$, and (3) the representative trajectory $\mathcal{C}_{\text{vis}}$. A structured prompt is constructed via a $\text{prompt}_{\text{gen}}$ function and passed to the MLLM:

$$L_{\text{new}} = \mathcal{M}_{\text{gen}}(I_0, I_t, L, \mathcal{C}_{\text{vis}}, \text{prompt}_{\text{gen}}), b_{\text{new}} = \mathcal{M}_{\text{ground}}(I_t, \text{prompt}_{\text{ground}}), \tag{8}$$

where $L_{\text{new}}$ is the updated textual query reflecting the target's current appearance or behaviour, and $b_{\text{new}}$ is a predicted box in frame $t$. The refined language $L_{\text{new}}$ is used in subsequent grounding and tracking steps, enabling the system to recover from drift and continue operating under challenging visual conditions, all in a training-free manner.

## 4 EXPERIMENT

### 4.1 EXPERIMENTAL SETUP

We conduct all experiments on a server equipped with 8 NVIDIA A800-SXM4-80GB GPUs. Our method integrates Qwen2-VL-72B (Wang et al., 2024a) as the multimodal large language model (MLLM), Grounding DINO-T (Liu et al., 2024a) as the grounding vision-language model (GVLM), and UVLTrack-B (Ma et al., 2024) as the base tracker (VLT). For modules involving confidence-based activation and MLLM interaction, we set the confidence threshold $\tau$ to 0.5 by default. The monitor window length $k$, used for determining when to trigger MLLM-based refinement, is set to 10 in most cases but may be adjusted for different datasets based on their tracking characteristics. In the text refinement module's experimental setup, the trajectory length N for representing the current tracking context is specifically set to 5 frames

### 4.2 COMPARISON WITH EXISTING TRACKERS

Table 1 summarizes the performance of our method and several state-of-the-art language-guided trackers under the language-only setting, where our approach consistently achieves the best results across all datasets and evaluation metrics. On **OTB99** (Li et al., 2017), which contains 48 short-term sequences with challenges such as occlusion, scale variation, and motion blur, our method reaches an AUC of 65.3%, precision of 88.2%, and normalized precision of 79.8%, clearly highlighting the effectiveness of our MLLM-based modules in enhancing short-term tracking accuracy. On the large-scale **TNL2K** (Wang et al., 2021) benchmark with 700 diverse test sequences spanning RGB, thermal, cartoon, and synthetic modalities, our tracker achieves an AUC of 58.2%, surpassing the second-best method by 3.3% and showing strong adaptability to heterogeneous scenarios. Finally, on the long-term benchmarks **LaSOT** (Fan et al., 2019) and **LaSOT$_{ext}$** (Fan et al., 2021), which

Table 2: Quantitative results of first-frame localization using AvgIoU across four benchmarks. VLT-Based methods represent conventional trackers, Detection-Based methods rely on open-set object detectors, while MLLM-Augmented methods leverage large multimodal models for localization.

| Method | Category | OTB99 | TNL2K | LaSOT | LaSOT$_{ext}$ |
|---|---|---|---|---|---|
| JointNLT | VLT-Based | 54.18 | 59.04 | 52.17 | 28.65 |
| UVLTrack-B | VLT-Based | 59.82 | 60.13 | 52.17 | 24.75 |
| GroundingDINO-Only | Detection-Based | 58.79 | 57.28 | 49.92 | 20.50 |
| QWEN2-VL-Only | MLLM-Augmented | 67.47 | 67.46 | 60.28 | 44.62 |
| MAGTrack-MGM | MLLM-Augmented | **71.36** | **70.18** | **64.09** | **46.84** |

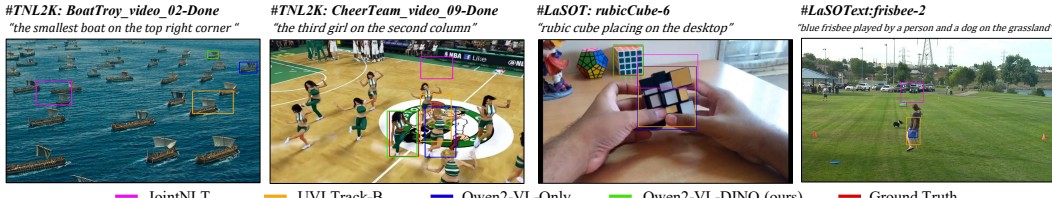

Figure 3: Illustrations of first-frame localization results highlighting differences between different methods in challenging scenarios with semantic ambiguity and visually similar distractors.

emphasize complex conditions and semantically rich queries, our method integrates dynamic text refinement and multimodal reasoning to alleviate drift and semantic mismatch, achieving clear improvements over prior methods, with especially large gains on LaSOT$_{ext}$ (+7.3% AUC and +12.7% P). These results collectively demonstrate the robustness and generalization ability of our framework across both short- and long-term, homogeneous and heterogeneous tracking scenarios.

## 4.3 ANALYSIS

**Robust First-Frame Grounding via MLLM-Guided Disambiguation.** Accurate target localization in the first frame is critical for robust language-guided visual tracking, especially in the absence of any initial bounding box and when only a natural language description is available. However, existing trackers such as UVLTrack (Ma et al., 2024) and JointNLT (Zhou et al., 2023) are not designed to perform localization from scratch based solely on language input. Instead, they typically rely on shallow vision-language matching mechanisms or fixed embedding comparisons, which are insufficient to resolve semantic ambiguities or to filter out visually similar distractors in complex scenes. Similarly, detection-based approaches like GroundingDINO (Liu et al., 2024a) can provide region proposals but lack the semantic reasoning ability to disambiguate complex queries. As shown in Table 2, these methods exhibit limited localization accuracy in the first frame.

In contrast, multimodal large language models (MLLMs) possess strong semantic reasoning capabilities, enabling them to interpret complex natural language queries and identify corresponding regions in an image. While this improves grounding in many cases, MLLMs alone can still fail in challenging scenarios. For example, as illustrated in Figure 3, given the query *"the smallest boat on the top right corner"*, multiple similar-looking boats may cause ambiguity and lead to inaccurate localization. To mitigate this, we incorporate explicit spatial priors from GroundingDINO into the MLLM pipeline: candidate regions are first proposed and then refined by semantic reasoning. This combination leverages the global language understanding of MLLMs and the spatial precision of visual grounding, resulting in improved robustness and localization accuracy across diverse cases.

**Adaptive Language Refinement for Dynamic Visual Semantics.** While initial localization sets the foundation for tracking, visual appearance and semantic relevance can drift over time, especially in long or dynamic sequences. To address this issue, we propose a Text Refinement module that adaptively updates the natural language description based on observed tracking history and current visual cues.

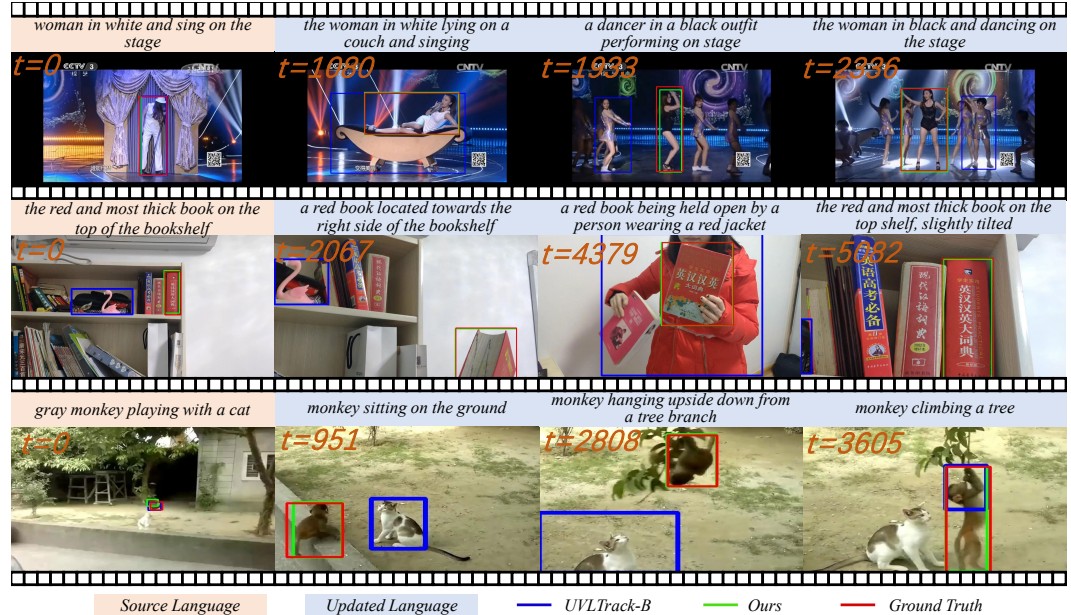

Figure 4: Visualization of text refinement and tracking under semantic variation. Each row corresponds to a video sequence and displays several keyframes sampled over time, along with the original language description, the refined language, and the tracking results from UVLTrack-B (Ma et al., 2024), our method, and the ground truth.

Table 3: Average image-text alignment scores using original and refined descriptions on updated frames, evaluated with CLIP (Radford et al., 2021) using ViT-B/32 and RN-50 as image encoders.

| Setting | TNL2K | LaSOT | LaSOT$_{ext}$ |
|---|---|---|---|
| RN50 (Original) | 0.1789 | 0.1916 | 0.1856 |
| **RN50 (Refined)** | **0.1849** | **0.1926** | **0.1861** |
| ViT-B/32 (Original) | 0.2421 | 0.2576 | 0.2470 |
| **ViT-B/32 (Refined)** | **0.2497** | **0.2590** | **0.2482** |

Table 4: Ablation study of the MLLM-based Grounding Module (MGM) and Text Refinement Module (TRM) on three datasets, showing only the AUC metric.

| Method | TNL2K | LaSOT | LaSOT$_{ext}$ |
|---|---|---|---|
| UVLTrack-B | 54.85 | 56.67 | 30.58 |
| + MGM | 56.56 | 59.03 | 40.18 |
| + TRM | **58.24** | **60.11** | **42.22** |
| JointNLT | 54.55 | 56.83 | 34.99 |
| + MGM | 54.84 | 57.38 | 37.44 |
| + TRM | **55.39** | **57.49** | **37.77** |

As shown in the qualitative examples in Figure 4, the original language often becomes semantically misaligned with the target object. For instance, in the first sequence, the description *"woman in white and sing on the stage"* becomes less accurate as the target transitions to lying on a couch, prompting an updated description: *"the woman in white lying on a couch and singing"*. These refined descriptions enable more accurate grounding by the MLLM and improve the robustness of tracking under significant appearance or context changes. We further validate the effectiveness of our Text Refinement module through image-text matching experiments. Specifically, we use the ground-truth bounding boxes to crop the target object from each frame where refinement was triggered, and compute the text-to-image alignment score using CLIP (Radford et al., 2021). Both the original and refined descriptions are encoded along with the cropped image region, and their similarity is measured via cosine distance in the CLIP embedding space. As reported in Table 3, the refined descriptions consistently yield higher alignment scores across three datasets, using both RN-50 (He et al., 2016) and ViT-B/32 (Dosovitskiy et al., 2020) as image encoders. These results indicate that the updated language more accurately reflects the visual content of the target, thereby improving the semantic grounding and reliability of the overall tracking process.

Table 5: Comparison of different MLLM backbones across four benchmarks. Localization performance is evaluated using AvgIoU, and tracking performance is reported using AUC.

| Model | First-Frame Localization (AvgIoU) | | | | Tracking (AUC) | | | |
|---|---|---|---|---|---|---|---|---|
| | OTB99 | TNL2K | LaSOT | LaSOT$_{ext}$ | OTB99 | TNL2K | LaSOT | LaSOT$_{ext}$ |
| Baseline | 59.82 | 60.13 | 52.17 | 24.75 | 60.09 | 54.85 | 56.67 | 30.58 |
| Qwen2-VL-7B | 68.47 | 63.79 | 59.49 | 43.68 | 65.58 | 55.40 | 59.87 | 41.88 |
| InternVL3-78B | 71.23 | 68.40 | 61.43 | 41.92 | 64.29 | 57.33 | 59.40 | 41.16 |
| Qwen2-VL-72B | 71.36 | 70.18 | 64.09 | 46.84 | 65.33 | 58.24 | 60.11 | 42.22 |

## 4.4 ABLATION STUDY

To better understand the role of each component, we perform ablation studies on TNL2K, LaSOT, and LaSOT$_{ext}$ (Table 4). Incorporating the MLLM-based Grounding Module (MGM) consistently improves both UVLTrack and JointNLT, confirming its plug-and-play nature and effectiveness in enhancing first-frame grounding with external visual cues. For instance, adding MGM to UVLTrack yields a gain of +1.7% AUC on TNL2K and +9.6% on LaSOT$_{ext}$, showing substantial improvements especially in challenging long-term scenarios. The Text Refinement Module (TRM) further boosts performance, yielding the best results across all datasets; e.g., UVLTrack achieves an additional +1.1% AUC on LaSOT with TRM, while JointNLT gains +0.3% on LaSOT$_{ext}$. These results confirm the complementary benefits of MGM and TRM: while MGM mainly contributes to spatial grounding accuracy, TRM enhances temporal robustness through adaptive language refinement. The largest improvement is observed on LaSOT$_{ext}$ (+11.6% over the UVLTrack baseline), highlighting the strong generalization of our design to long-term and unseen scenarios. Overall, the consistent gains across backbones and datasets demonstrate that our modules not only provide cumulative benefits but also transfer seamlessly across diverse tracker architectures, making them broadly applicable for future vision-language tracking frameworks.

## 4.5 SCALABILITY AND EFFICIENCY WITH VARYING MLLM BACKBONES

In addition to component-level analysis, we further examine the influence of the underlying MLLM used in our framework. As detailed in Table 5, we compare several large multimodal models with varying architectures and parameter scales, including Qwen2-7B (Wang et al., 2024a), Qwen2-72B (Wang et al., 2024a), and InternVL3-78B (Zhu et al., 2025). Our method achieves consistently strong performance across all backbones, confirming its robustness to the choice of MLLM. Although the 72B-scale models yield the best results overall, the 7B variant performs competitively while offering significantly lower inference latency and computational cost. This demonstrates that our framework is not only effective but also flexible and practical for real-world deployment scenarios, where resource constraints may limit the use of extremely large models. Detailed analysis of inference speed can be found in Appendix A.2.

## 5 CONCULSION

In this work, we propose MAGTrack, a flexible framework powered by Multimodal Large Language Models (MLLMs) for language-guided visual tracking. Without requiring any additional training, MAGTrack incorporates two plug-and-play components: an MLLM-based Grounding Module (MGM) that improves initial localization from ambiguous descriptions, and an MLLM-based Text Refinement Module (TRM) that adaptively updates the language query based on visual context during tracking. Together, these components enable seamless integration with existing trackers and enhance tracking robustness. Extensive experiments across multiple benchmarks demonstrate consistent improvements, highlighting the potential of MLLMs as powerful reasoning engines for vision-language tracking tasks. However, the inference cost of large MLLMs may limit their applicability in real-time or resource-constrained settings. Future work may explore lightweight alternatives or distillation-based approaches to improve efficiency. Our framework opens up promising directions for natural human-computer interaction, including multimodal surveillance, robotics, and assistive systems, where flexible language-based control and adaptation are essential.

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

# A APPENDIX

## A.1 USE OF LARGE LANGUAGE MODELS (LLMS)

In this work, Large Language Models (LLMs) were employed solely as writing assistants for language polishing, including improving grammar, clarity, and readability. They were not used for generating, analyzing, or interpreting research data, nor for deriving any scientific conclusions. All core ideas, methods, and experimental results presented in this paper are entirely the work of the authors.

## A.2 LIMITATIONS

While our method demonstrates strong performance across four benchmarks without additional training, it relies on large-scale multimodal language models (MLLMs), such as Qwen2-VL-72B, which can be computationally expensive to deploy in real-time or resource-constrained environments. To evaluate the runtime performance of our method, we conduct all testing experiments on an NVIDIA A800-SXM4-80GB GPU. Although our framework is designed to adaptively trigger text-based re-grounding based on the confidence scores from the visual tracker, we adopt a fixed-interval re-grounding strategy in this experiment to better isolate and measure the computational overhead introduced by MLLMs. Specifically, we simulate the process of query updating by performing re-grounding every 200 frames, which approximates a moderate update frequency in typical video scenarios where the target's appearance changes gradually over time. As shown in Table 6, the baseline tracker (UVLTrack-B) runs at 48.93 FPS. Incorporating MLLMs introduces notable latency: the 7B variant (e.g., Qwen2-VL-Chat) achieves 17.04 FPS, while the 72B model (Qwen2-VL-72B) further reduces speed to 7.67 FPS due to its significantly larger model size and visual-textual reasoning cost. Despite the runtime drop, MLLMs substantially enhance grounding and text refinement, particularly in challenging scenes with ambiguous language or drastic appearance changes. Future work could explore lightweight alternatives, such as distilled or quantized MLLMs, and dynamic scheduling strategies that invoke re-grounding based on tracking confidence rather than fixed intervals. These improvements may reduce latency while maintaining robustness across diverse scenarios.

Table 6: Runtime speed (FPS) comparison of different tracking methods.

| Metric | UVLTrack-B | MAGTrack-7B (Ours) | MAGTrack-72B(Ours) |
|--------|------------|--------------------|--------------------|
| FPS | 48.93 | 17.04 | 7.67 |

## A.3 BROADER IMPACTS

While the core contribution of this work lies in proposing a training-free and modular framework for language-guided object tracking, its practical value emerges in enabling user-friendly interaction—allowing tracking through natural language instead of manual annotations. This lowers the barrier for real-world applications such as assistive systems or human-computer interfaces. On the downside, the use of large vision-language models introduces efficiency and resource constraints, which may hinder deployment in time-sensitive or resource-limited environments.

## A.4 PROMPT FOR LLM

n this section, we present the prompt designs used for three core tasks involving large language models: grounding generation, box selection, and textual description refinement. The corresponding examples are illustrated in Figure 5, Figure 6, and Figure 7, respectively. For clarity, the input components provided to the model are highlighted in blue.

## A.5 REPRESENTATIVE FRAME SAMPLE STRATEGY

To reduce the number of frames passed into the MLLM while preserving temporal diversity, we design a simple yet effective strategy to select representative frames based on frame indices. Specifically, given the target frame index, we divide the timeline into two parts: the early segment and the tail segment. For the early part, we sample $k_{main}$ frames at fixed ratio intervals between 10% and 60%

---

**Prompt for Visual Grounding**

Detect the bounding box of {description}. You must answer only in the following format:
    <|object_ref_start|>...<|object_ref_end|><|box_start|>(x1,y1),(x2,y2)<|box_end|>
    If uncertain, still provide your best guess. An answer in the required format is always mandatory.

---

Figure 5: Prompt for Visul Grounding.

---

**Prompt for Box Selection**

Here are {num} boxes: {boxes_list}. Which box best matches the phrase: "{language}"? Please output in the following strict format:
    <answer>...your final description [x1,y1,x2,y2] here...</answer>
    The answer must be one of the boxes from the list.

---

Figure 6: Prompt used for box selection based on language queries.

of the target frame, selecting the closest available frame to each target ratio. For the tail part, we select the last two available frames from the range between 80% and 100% of the target frame. All selected frames are then deduplicated and sorted. This strategy ensures that the sampled frames capture both long-range and short-range temporal context while maintaining efficiency, as detailed in **Algorithm 1**.

---

**Algorithm 1** Select Representative Frames

---

**Require:** Frame index list $F$, target frame $t$, number of frames $k$
**Ensure:** A set of $k$ representative frames
  1: **if** $|F| \leq k$ **then**
  2:     **return** all frames in $F$
  3: **end if**
  4: Divide $k$ as $k = k_{\text{main}} + 2$
  5: Compute $k_{\text{main}}$ ratio points in $[0.1, 0.6]$
  6: **for** each ratio $r$ **do**
  7:     $f_r \leftarrow$ frame closest to $r \cdot t$
  8:     Add $f_r$ to selected set
  9: **end for**
 10: From frames in $[0.8t, t]$, select last two
 11: Add them to selected set
 12: Remove duplicates and sort
 13: **return** selected frames

---

---

**Prompt for Language Generation**

**You are given the following information from the same video:**

1. The **source frame** (frame 0) provides a textual description and bounding box of an object.

2. The **target frame** is a later frame (frame {curr_id}) where the object may have changed location, appearance, or status.

3. Optionally, several intermediate points track the object's center as it moves between the two frames.

**Your task is to:**

1. Analyze the position and visual cues of the object in the source frame.

2. Use the intermediate trajectory to infer the likely location of the object in the target frame.

3. Generate a new textual description that best describes the object in the target frame, based on its new appearance, pose, or behavior.

4. Provide the bounding box in the target frame.

**Input:**

1. {trajectory_dict}

2. <|src_description_start|>{src_language}<|src_description_end|>

3. <|src_box_start|>{src_coor}<|src_box_end|>

**Always follow this format in your answer:**

<|think|>...your reasoning about how the object moved and what it looks like now...<|think|>

<|tgt_description_start|>...updated textual description...<|tgt_description_end|>

<|tgt_box_start|>(x1, y1), (x2, y2)<|tgt_box_end|>

---

Figure 7: Prompt used for language and box generation across frames, based on object trajectory.

