# OpenReview forum: "MAGTrack: MLLM-Augmented Grounding and Text Refinement for Language-Guided Tracking"
_ICLR.cc/2026/Conference — ICLR 2026 Conference Withdrawn Submission_

### Official Review · Reviewer_iXjB · 2025-10-27

**Soundness:** 2
**Presentation:** 2
**Contribution:** 2
**Rating:** 2
**Confidence:** 5

**Summary:**

This paper presents MAGTrack, a training-free framework for language-guided visual tracking that integrates Multimodal Large Language Models (MLLMs) through two plug-and-play components: the MLLM-based Grounding Module (MGM) and the MLLM-based Text Refinement Module (TRM). The proposed method leverages the reasoning and generative capabilities of MLLMs to address key challenges in the language-only tracking setting—namely, initial grounding ambiguity and temporal mismatch between static text and dynamic visual content. Experiments on four benchmarks (OTB99, TNL2K, LaSOT, and LaSOText) show consistent improvements over existing approaches.

**Strengths:**

- The proposed MGM and TRM are plug-and-play modules that can be seamlessly integrated into existing vision-language trackers (e.g., UVLTrack, JointNLT), demonstrating good generality and transferability.
- The paper is easy to follow.

**Weaknesses:**

- The proposed method appears to be mainly a combination of existing MLLM-based reasoning modules and conventional tracking models, making it more of an engineering exploration rather than a fundamentally new algorithmic contribution.
- It is recommended to compare with more recent state-of-the-art methods, such as DUTrack and other strong baselines, to better highlight the advantages of the proposed framework.
- The paper should include more detailed ablation experiments, especially regarding confidence thresholds, update frequencies, and frame sampling strategies, to analyze the sensitivity and robustness of key design choices.
- Since the integration of MLLMs into tracking has already been explored in several recent works, the authors are encouraged to restate their motivation more clearly and emphasize the specific novelty or unique perspective of their approach to improve overall clarity and contribution.

**Questions:**

See Weaknesses.

---

### Official Review · Reviewer_3Zi9 · 2025-10-29

**Soundness:** 2
**Presentation:** 2
**Contribution:** 2
**Rating:** 2
**Confidence:** 5

**Summary:**

This paper introduces MAGTrack, a modular framework for language-guided object tracking that integrates multimodal large language models (MLLMs) through two plug-and-play components: an MLLM-based Grounding Module (MGM) for initial localization, and an MLLM-based Text Refinement Module (TRM) for dynamically updating textual descriptions during tracking. The method aims to perform training-free, language-only tracking by combining reasoning from large vision-language models with conventional visual trackers. Experiments are conducted on four benchmarks (OTB99, TNL2K, LaSOT, LaSOText). While the paper is well written and the motivation is clear, I find that the overall novelty is limited, the efficiency is poor, and the empirical validation is insufficient for a top conference like ICLR.

**Strengths:**

1. The paper is well written and the problem of language-only tracking is clearly motivated.
2. The integration of Multimodal Large Language Models (MLLMs) into language-assisted tracking is indeed an emerging and promising research direction.
3. The idea of using MLLMs as plug-and-play components (MGM, TRM) is simple and easy to integrate with existing trackers without a training process.

**Weaknesses:**

1. Low novelty and shallow contribution. Both modules (MGM and TRM) are straightforward combinations of Grounding DINO + MLLM reasoning and text rewriting via prompting. There is no new algorithm, learning strategy, or theoretical insight. The framework is more of an engineering integration than a research innovation.
2. Extremely high computational cost and low efficiency. Tracking is inherently a real-time task — models that cannot operate at interactive or near-real-time speeds (≥20 FPS) are difficult to justify, regardless of modest accuracy improvements. The reported speed of only 7–17 FPS with large MLLMs (Qwen2-VL-72B/7B) indicates that the proposed framework is far from deployable and serves mainly as a conceptual prototype. The reliance on Qwen2-VL-72B/7B makes deployment impractical, and there are no lightweight or efficient variants.
3. Incomplete and weak experimental comparison. Some existing methods like MambaVLT[1] and UVLTrack-L[2] are missed in the Table 1.
4. Limited task scope. A major conceptual limitation of the paper is that it restricts itself to language-only tracking, despite using UVLTrack as the base tracker—a model initially designed to unify language-guided (NL) and box-initialized (NL+BBOX) tracking within a single framework. Given that MAGTrack claims to be training-free and modular, it is unclear why the authors did not extend the framework to both paradigms, or at least demonstrate compatibility with the UVLTrack’s dual-mode capability. This narrow focus on the language-only setting makes the contribution appear incomplete and less general than prior unified methods such as UVLTrack or MambaVLT.
[1] Liu X, Zhou L, Zhou Z, et al. Mambavlt: Time-evolving multimodal state space model for vision-language tracking[C]//Proceedings of the Computer Vision and Pattern Recognition Conference. 2025: 8731-8741.
[2] Ma Y, Tang Y, Yang W, et al. Unifying visual and vision-language tracking via contrastive learning[C]//Proceedings of the AAAI Conference on Artificial Intelligence. 2024, 38(5): 4107-4116.

**Questions:**

1. Since MAGTrack builds directly upon UVLTrack, which already supports both language-only and language-box tracking, why is the proposed framework evaluated exclusively on the language-only setting? Wouldn’t a unified, training-free framework be more meaningful and aligned with the original UVLTrack motivation?
2. Have you considered smaller MLLMs or distilled models to achieve real-time tracking?
3. How sensitive is the system to the prompt templates and trigger threshold? Is performance stable across different prompt wordings and confidence thresholds?

---

### Official Review · Reviewer_1wzG · 2025-10-30

**Soundness:** 2
**Presentation:** 3
**Contribution:** 2
**Rating:** 4
**Confidence:** 5

**Summary:**

This paper proposes MAGTrack, a framework for language-guided object tracking that integrates Multimodal Large Language Models (MLLMs) to address key challenges in this domain. The framework consists of two plug-and-play modules, where MLLM-based Grounding Module (MGM) enhances first-frame localization by leveraging MLLM reasoning to handle semantically ambiguous or visually similar objects; and MLLM-based Text Refinement Module (TRM) dynamically updates the initial natural language description during tracking, maintaining alignment between language and visual content.

**Strengths:**

- Innovative Use of MLLMs.

The framework leverages the reasoning and generative capabilities of MLLMs in a novel way, addressing challenges like semantic ambiguity and temporal misalignment in language-guided tracking.

- Dynamic Text Refinement.

The TRM module is a strong addition, allowing the tracker to adapt to appearance changes and contextual variations, which is a critical limitation in existing methods.

- Modular Design.

MAGTrack is designed as a training-free, plug-and-play framework, making it theoretically compatible with various existing trackers without requiring retraining.

**Weaknesses:**

- Lack of Novelty.

While the integration of MLLMs is interesting, the overall framework does not introduce fundamentally new ideas compared to existing vision-language tracking (VLT) methods. Many components, such as the use of grounding models and language refinement, are incremental rather than groundbreaking. The framework feels like an extension of current VLT methods, such as UVLTrack or JointNLT, rather than a significant departure from them.
Insufficient

- Experimental Comparison.

The paper avoids direct comparisons with some of the strong VLT baselines, such as  VLT. This omission raises questions about whether the improvements claimed are genuinely significant. Including these comparisons would provide a more comprehensive evaluation of MAGTrack's performance relative to state-of-the-art methods.

- Limited Practical Applicability.

The reliance on large-scale MLLMs severely limits the framework's real-world applicability. Even with the smallest model used (7B parameters), the framework cannot achieve real-time performance, which is a critical requirement for tracking tasks. The reported speed of 17 FPS for the 7B model and 7.67 FPS for the 72B model highlights this issue. Given the high computational cost of MLLMs, the framework is unlikely to be practical for real-time or resource-constrained environments, which limits its utility in real-world applications such as robotics or surveillance.

**Questions:**

see above

---

### Note · Authors · 2025-11-14

I have read and agree with the venue's withdrawal policy on behalf of myself and my co-authors.